# Post-Analysis of Predictive Modeling with an Epidemiological Example

**DOI:** 10.3390/healthcare9070792

**Published:** 2021-06-24

**Authors:** Christina Brester, Ari Voutilainen, Tomi-Pekka Tuomainen, Jussi Kauhanen, Mikko Kolehmainen

**Affiliations:** 1Department of Environmental and Biological Sciences, University of Eastern Finland, Yliopistonranta 1 E, P.O. Box 1627, FI-70211 Kuopio, Finland; mikko.kolehmainen@uef.fi; 2Institute of Public Health and Clinical Nutrition, University of Eastern Finland, Yliopistonranta 1 C, P.O. Box 1627, FI-70211 Kuopio, Finland; ari.voutilainen@uef.fi (A.V.); tomi-pekka.tuomainen@uef.fi (T.-P.T.); jussi.kauhanen@uef.fi (J.K.)

**Keywords:** post-analysis of data-driven models, rule design, multi-objective optimization, model performance, prediction of cardiovascular death

## Abstract

Post-analysis of predictive models fosters their application in practice, as domain experts want to understand the logic behind them. In epidemiology, methods explaining sophisticated models facilitate the usage of up-to-date tools, especially in the high-dimensional predictor space. Investigating how model performance varies for subjects with different conditions is one of the important parts of post-analysis. This paper presents a model-independent approach for post-analysis, aiming to reveal those subjects’ conditions that lead to low or high model performance, compared to the average level on the whole sample. Conditions of interest are presented in the form of rules generated by a multi-objective evolutionary algorithm (MOGA). In this study, Lasso logistic regression (LLR) was trained to predict cardiovascular death by 2016 using the data from the 1984–1989 examination within the Kuopio Ischemic Heart Disease Risk Factor Study (KIHD), which contained 2682 subjects and 950 preselected predictors. After 50 independent runs of five-fold cross-validation, the model performance collected for each subject was used to generate rules describing “easy” and “difficult” cases. LLR with 61 selected predictors, on average, achieved 72.53% accuracy on the whole sample. However, during post-analysis, three categories of subjects were discovered: “Easy” cases with an LLR accuracy of 95.84%, “difficult” cases with an LLR accuracy of 48.11%, and the remaining cases with an LLR accuracy of 71.00%. Moreover, the rule analysis showed that medication was one of the main confusing factors that led to lower model performance. The proposed approach provides insightful information about subjects’ conditions that complicate predictive modeling.

## 1. Introduction

The increasing volume of data collected and the expanding computational resources dictate current trends in data-driven modeling [1]. It is no longer surprising that models outperform human experts in many areas [2]. Yet this high performance goes hand-in-hand with significant growth in model complexity. This tendency results in greater intricacy of the use of data-driven models in the medical domain, where model interpretability is of primary importance [3]. While predicting diseases, one may even choose to use less accurate “white box” models (easily interpretable with a simple structure), such as decision trees or rules, rather than to unpack “black box” models (hardly interpretable with a complex structure) [4,5].

To address the growing complexity of data-driven models and to understand the non-trivial logic behind their decisions, a number of methods for post-analysis have been proposed recently [6]. Some of them apply local approximations with simpler but interpretable models [7], while others estimate feature importance using permutation techniques [8]; additionally, they search for influential samples that greatly affect model parameters [9]. Alternatively, there are methods applied while generating a model that aim to find a trade-off between model accuracy and complexity [10]. At this stage, optimal sampling techniques or model structures that lead to higher accuracy and lower complexity might be determined [11,12].

In this study, we addressed a particular question of model post-analysis, i.e., the conditions in the sample that lead to higher or lower model performance. To answer this question in the low-dimensional predictor space, error heatmaps can be used, since they allow the visualization and identification of subpopulations with high and low error rates considering one or two predictors [13]. Then, decision trees inherently perform error analysis, as their terminal nodes contain subpopulations—whose characteristics are discovered by moving up the tree—and model performance is easily estimated for each terminal node [14]. Moreover, stratification is widely used when comparing the model performance of different subpopulations (men vs. women, young vs. old, with vs. without a particular condition), which may positively affect model predictive ability compared to training on combined samples [15]. However, these approaches are not suitable for multivariate error analysis in the high-dimensional predictor space. Alternatively, an “unreliability” score, proposed by Myers et al. [16], is individually applied for each subject, and its high values indicate decreased model accuracy. This approach requires additional analysis of cases with high unreliability to extract knowledge about “difficult” subjects.

Taking into account the limitations of the existing studies, in the presented epidemiological example we revealed “easy” and “difficult” cases for the model when it operated in the high-dimensional predictor space. Particularly, we trained Lasso logistic regression (LLR) to predict cardiovascular death using data from the Kuopio Ischemic Heart Disease Risk Factor Study (KIHD) [17]. After validating the model, we generated a set of compact rules that covered samples with extremely high or low model performance. Generally speaking, this paper presents the model-independent approach for post-analysis and shows that this approach not only reveals when the model predictions are less reliable, but also finds out from which perspectives the model requires improvement and what kind of new samples might be collected to bring more information into “difficult” regions of the predictor space. Although we did not have a specific hypothesis to test, this exploratory post-analysis yielded interesting findings.

## 2. Materials and Methods

### 2.1. KIHD: Baseline Cohort

The data utilized in the current study were collected in 1984–1989 in the city of Kuopio and the surrounding area in Eastern Finland, whose population was recorded to have one of the highest rates of coronary heart disease [17,18]. The KIHD study is an ongoing project, where the study outcomes are derived from the national registers annually.

The baseline examinations (1984–1989) comprised 2682 randomly selected middle-aged men (42–60 years old), whose health state was carefully described with thousands of physiological, clinical, biochemical, psychological, and socioeconomic measurements. As a starting point, 950 predictors were preselected by the domain expert to perform predictive modeling. In our experiments, the outcome variable was “death from a cardiovascular disease by 2016” referring to codes I00–I99 of the 10th International Classification of Diseases (ICD 10) [19]. Before training a predictive model, preprocessing was applied (Figure 1). Every time we trained the model, the predictors were normalized to the interval (0, 1) using the scaler fitted on the training data.

### 2.2. Multi-Objective Rule Design for the Model Post-Analysis

Let us consider a dataset of *n* subjects described with *m* predictors each: X={x(1),x(2), …, x(n)}, where x(i)∈ℝm, i=1,n¯, and an outcome variable: y={y(1),y(2),…,y(n)}, where y(i)∈{0, 1}. In the context of epidemiological data, y(i)=0 and y(i)=1 designate “negative” and “positive” regarding a diagnosis. The sample (X, y)={(x(1),y(1)), (x(2),y(2)), …,(x(n),y(n))} is used to train a model that performs mapping f:x(i)⟼y(i), i.e., predicts a value of y(i) for a given vector of predictors x(i).

In this study, we applied an LLR model, since in our previous experiments, it demonstrated the highest performance on the KIHD data [21]. LLR describes a relationship between a linear combination of predictors and the probability of having a disease in the form of a sigmoid function, P(y=1|x)=11+exp(−(ω0+ω1x1+…+ωmxm)), as a traditional logistic regression model does [22]. However, LLR is a penalized regression, whose cost function includes an L1-regularization term ${∥\omega ∥}_{1}$ in addition to the cross-entropy error $\sum_{i=1}^{n}\log\left(1+\exp\left(-y^{\left(i\right)}\cdot \langle \omega ,x^{\left(i\right)}\rangle \right)\right)$ [23]:(1)$$\underset{\omega }{\min\;}{∥\omega ∥}_{1}+C\cdot \overset{n}{\underset{i=1}{\sum }}\log\left(1+\exp\left(-y^{\left(i\right)}\cdot \langle \omega ,x^{\left(i\right)}\rangle \right)\right)$$
where C defines a shrinkage parameter (the regularization amount) equal to 0.15 in this study.

If $P(y^{\left(i\right)}=1|x^{\left(i\right)})\geq \alpha$, then, $y^{\left(i\right)}=1$; otherwise, $y^{\left(i\right)}=0$. For the imbalanced sample, a cutoff value α requires adjustment (this typically equals 0.5 for the balanced sample); therefore, α is defined as a proportion of “positive” cases in the training sample.

To estimate the model performance on the test data, we executed 50 independent runs of a k-fold cross-validation with stratification, where k= 5. LLR was implemented using the scikit-learn library [24]. In each run, we collected the true values of $y^{\left(i\right)}$ and the model outcomes on the test data to calculate the number of times the predictions were correct and wrong for each subject. Then, for “positive” cases, we produced the amount of true positive (TP) and false negative (FN) predictions: $TP_{i}$ and $FN_{i}$; for “negative” cases, we calculated the number of true negative (TN) and false positive (FP) predictions: $TN_{i}$ and $FP_{i}$. This model evaluation served as a basis for further analysis, aiming to reveal groups of “easy” and “difficult” cases.

Usually, the model performance is described with statistics averaged over the whole sample. These aggregated values are helpful when comparing models, but they do not reflect the distribution of errors in the space of predictors. In other words, the model performance varies for different subjects and we might be more or less confident in the correctness of predictions depending on the subject’s characteristics. Therefore, we propose an approach to automatically generate a set of rules that represent the conditions leading to higher or lower model performance compared to the average level.

Let rule denote a set of concurrent conditions “$x_{j}$ equals $a_{j}$”:(2)$$rule=\left(x_{1}=a_{1}\right)\;\mathrm{and}\;\left(x_{2}=a_{2}\right)\ldots \mathrm{and}\;\left(x_{j}=a_{j}\right)\;\mathrm{and}\ldots ={⋂}_{j\in J}(x_{j}=a_{j}),$$
where $a_{j}$ is a particular level (category) of a predictor $x_{j}$ and J contains indices of predictors included in the rule. To define levels $a_{j}$, we first need to introduce categories for each predictor $x_{j}$ based on its distribution. For dichotomous variables, as well as for ordinal or continuous variables with fewer than eight different values in the sample, i.e., $\left|x_{j}\right|<8$: $a_{j}\in \left\{0,\;1,\;\ldots ,\max\left(x_{j}\right)\right\}$. For ordinal or continuous variables with eight and more possible values, i.e., $\left|x_{j}\right|\geq 8$, we introduce $N_{bins}$ intervals using the *k*-means clustering method, $N_{bins}=4$. Each interval corresponds to a particular level of $x_{j}$: $\left[x_{j}{}^{lower_{0}},x_{j}{}^{upper_{0}}\right]\to 0,\;\ldots ,\;\left[x_{j}{}^{lower_{\left(N_{bins}-1\right)}},x_{j}{}^{upper_{\left(N_{bins}-1\right)}}\right]\to (N_{bins}-1)$.

Then, to generate rules of interest, i.e., to select predictors $x_{j}$ and to define their levels $a_{j}$, we solve two three-objective optimization problems:

Problem 1 (3): To define rules that describe subgroups of subjects with the maximum true positive rate (TPR) and true negative rate (TNR) (i.e., “easy cases”):(3)$$\{\begin{array}{c} TPR\left(rule\right)\to \max\;\\ TNR\left(rule\right)\to \max\\ N_{subjects}\left(rule\right)\to \max \end{array}$$

Problem 2 (4): To define rules that describe subgroups of subjects with the minimum TPR and TNR (i.e., “difficult cases”):(4)$$\{\begin{array}{c} TPR\left(rule\right)\to \min\;\\ TNR\left(rule\right)\to \min\\ N_{subjects}\left(rule\right)\to \max \end{array}$$

In both problems, the third criterion is used to maximize the number of subjects covered by the rule, which aims to find as general a pattern as possible.

To solve the multi-objective Problems (3) and (4), we apply the Non-dominated Sorting Genetic Algorithm III (NSGA III), which is a stochastic optimization algorithm operating with a population of solutions whose quality improves during the search [25]. NSGA III is based on a Pareto-dominance idea and returns a set of nondominated solutions (in our case, a set of rules), of which one cannot be preferred over another. Each solution in the population is coded with a binary string, a so called “chromosome”, and genetic operators such as selection, crossover, and mutation are applied to binary strings so that new solutions with better values of objective criteria are produced. Figure 2 explains how a binary coding is used to represent rules (2):

The number of bits $n_{j}^{bits}\;$ to code $a_{j}$ depends on the amount of levels $n_{j}^{levels}$ introduced for $x_{j}$, plus one additional level, meaning the absence of $x_{j}$ in the rule: $n_{j}^{bits}=\lceil \log_{2}\left(n_{j}^{levels}+1\right)\rceil$. If, after decoding, $a_{j}>\left(n_{j}^{levels}+1\right)$, this also means the absence of $x_{j}$ in the rule.

In this study, we used NSGA III implemented in the Platypus package [26]. Table 1 contains the algorithm settings.

Given the stochastic nature of NSGA III, we ran the algorithm 25 times for each problem, i.e., Problem 1 and 2 (3, 4), and then combined the final populations from all of the runs. Figure 3 summarizes the pipeline described; the source code might be found on GitHub [27].

In general, any multi-objective evolutionary algorithm (MOGA) might be applied in the proposed approach (Figure 3); therefore, the overall time complexity depends on the optimization algorithm used. Time complexities of different MOGAs are discussed in the article by Curry and Dagli [28].

Lastly, from all generated rules that excessively describe subgroups of different sizes with various levels of TPR and TNR, we selected the final set of rules meeting the following criteria:There are at least Supp subjects covered by the rule (the minimum rule support): Supp = 30.The difference between TPR and TNR does not exceed γ·max(TPR(rule), TNR(rule))): γ=0.1 (the rule is equally valid for “positive” and “negative” cases).The average model accuracy for subjects covered by the rule is either lower than $\alpha_{diff}$ or higher than $\alpha_{easy}$ (to define “difficult” and “easy” cases correspondingly): $\alpha_{diff}=50\%$, $\alpha_{easy}=95\%$.

From a practical point of view, the final set of rules allowed us to reveal the conditions (subjects’ features) that lead to higher or lower model performance, compared to the average level. First, this knowledge is helpful for revising the sample and collecting new data. Second, using these rules, we can introduce four categories of subjects: “easy” cases that are covered only by the rules with $Accuracy\left(rule\right)\geq \alpha_{easy}$; “difficult” cases that are covered only by the rules with $Accuracy\left(rule\right)\leq \alpha_{diff}$; “ambiguous” cases that are covered by the rules with $Accuracy\left(rule\right)\geq \alpha_{easy}$ and the rules with $Accuracy\left(rule\right)\leq \alpha_{diff}$; “not covered cases” that are covered by none of the rules. Applying the final set of rules to unseen data and categorizing samples in such a way provides us with additional information about the probability of wrong predictions and allows us to be more or less confident in the model outcome.

## 3. Results

First, we trained the LLR model predicting cardiovascular death in 50 runs of five-fold cross-validation and estimated the model performance on the test data: accuracy = 72.527%, TPR = 72.485%, and TNR = 72.552%. Despite splitting the data into the training and test samples randomly, for some subjects the correctness of the model predictions did not vary across the multiple runs: the model outcome was always (or in most of the runs) either right or wrong. To generate rules that would reveal the conditions of “easy” and “difficult” cases, we used a preprocessed set of predictors: we filtered out predictors that were not selected by LLR in at least one run. Thus, in the further analysis, 191 predictors were involved.

Next, after 25 independent runs of NSGA III, we ended up with 4355 rules for Problem 1 (3) and 4583 rules for Problem 2 (4). Appendix A presents these initial sets of rules in the criterion space TPR–TNR.

Then, we selected the final set of rules using criteria 1–3 from Section 2.2 and, as a result, we obtained 43 rules representing “easy” cases, for which the model accuracy was higher than $\alpha_{easy}=95\%,$ and 39 rules that represent “difficult” cases, for which the model accuracy was lower than $\alpha_{diff}=50\%$. These particular threshold values were chosen to reveal the subjects’ characteristics that lead to the extremely high and extremely low model accuracy, provided that the accuracy for the remaining “ambiguous” and “non-covered” cases is close to the model accuracy on the whole sample. We also aimed to have the number of “non-covered” cases along with “ambiguous” cases at least lower than half of the sample. Thus, the selected 82 rules divided the KIHD sample into four categories, whose characteristics, with respect to the model accuracy and the number of subjects, are given in Table 2. Moreover, to support our choice, a description of subjects’ categories obtained for other $\alpha_{easy}$ and $\alpha_{diff}$ is presented in the Appendix A.

To visualize the groups of subjects covered by the final set of rules, we introduced an 82-dimentional rule space, where each dimension defined whether a subject was covered by the rule or not. These binary vectors were used by t-SNE (t-Distributed Stochastic Neighbor Embedding) to project subjects into the two-dimensional space [29]. Figure 4 illustrates the “easy”, “difficult”, and “ambiguous” cases and how the model accuracy changes for different clusters of subjects. These clusters mean that, within two large groups of “easy” and “difficult” cases, there are different conditions leading to an increase or decrease in model performance.

In general, the results can be analyzed from two perspectives. First, we may pay attention to the rules that describe a particular subject and make conclusions at the subject level about the predictors that make it “easy” or “difficult” for the model. Second, we may carry out analysis at the rule level by reviewing the combinations of predictors and their values that commonly lead to “easy” or “difficult” cases. For example, the two following rules describe “difficult” cases:

(AMIHIST = no) *and* (BLPRESMED = yes) *and* (DIUR = yes) *and* (ISCHAEMIA = no),
(5)$$N_{subjects}=38,\;TPR=45.18\%\;\mathrm{and}\;TNR=42.95\%,\;$$
where AMIHIST is “Myocardial infarction in the past”, BLPRESMED is “Drug for blood pressure in last 7 days”, DIUR is “Diuretics”, and ISCHAEMIA is “Ischemia in exercise stress test”.

(6)$$(\mathrm{SMOHIST}=\mathrm{no})\;\mathrm{and}\;(\mathrm{PRESCRIP}=\mathrm{yes})\;\mathrm{and}\;(\mathrm{WAGE}=\mathrm{yes})\;\mathrm{and}\;(54\;\leq \;\mathrm{AGE}\;\leq \;55),\;N_{subjects}=86,\;TPR=48.00\%\;\mathrm{and}\;TNR=45.45\%,$$
where SMOHIST is “Smoking history, ever,” PRESCRIP is “Drugs prescribed by a doctor in last 7 days”, WAGE is “Salary: wages (if retired, then no)”, and AGE is “Age, years”.

What is interesting is that, in both groups, the subjects took medication. However, the first group had neither myocardial infarction in the past nor ischemia in exercise stress test, i.e., rule (5), and the second group had never smoked, i.e., rule (6). This implies that, in these rules, medication works as a factor that confuses the model for both the “cvd” and “no cvd” groups.

Lastly, to generalize the analysis of predictors and to extract the most important ones from the whole final set of rules, for each predictor, we calculated the number of unique “easy”, “difficult”, and “ambiguous” subjects covered by the rules that contain this predictor. Figure 5 presents the 50 most important predictors and their levels from the final set of rules. As can be noted, the same predictors with the same values might be involved in the rules that describe “easy” and “difficult” cases. For example, “ISCHAEMIA = no” in most of the cases corresponds to “easy” subjects (11.41% of the whole sample), but it also relates to “difficult” subjects (3.68% of the whole sample). On the contrary, the predictors “PRESCRIP = no” and “PGLINHIB = no” refer to “easy” subjects, whereas “PRESCRIP = yes” and “PGLINHIB = yes” correspond to “difficult” subjects. Moreover, “SELFMED = no” and “PGLIMED = no” only describe “easy” subjects.

Thus, these results support our previous conclusion about the role of medication in model performance: Its presence adversely affects both TPR and TNR [32].

## 4. Discussion

Typically, model performance is evaluated using the whole sample, which gives its average estimate. Yet it remains unclear for which samples the model is prone to making right predictions and for which samples its predictions are likely to be wrong. The high-dimensional predictor space commonly makes the analysis of results even more complicated. Therefore, in this study, we proposed an approach that enabled us to generate a set of rules that explain which samples were “easy” (predictions were more accurate) or “difficult” (predictions were less accurate) for the LLR model, trained and tested on the high-dimensional epidemiological data.

Since the average level of accuracy achieved by LLR was only 72.5% when predicting cardiovascular death for subjects from the KIHD cohort, the additional post-analysis was aimed at revealing those subjects’ features that lead to high or low model performance. First, the knowledge about “difficult” subjects might be helpful for revising the sample, as they are the first candidates for double-checking. Moreover, collecting new samples, if it takes place, with characteristics corresponding to “difficult” cases might increase the model performance by bringing more information to the poorly modeled areas of the predictor space. Then, applying the set of rules to unseen data provides more confidence in the model predictions if new samples are categorized as “easy” cases. Knowing the weak spots of the model is of the utmost importance for clinical applications, where additional tests should be performed for “difficult” cases to avoid wrong predictions.

Moreover, as a well-interpretable tool, rules explicitly report the logic behind decision making while analyzing each subject, which is especially valuable for medicine. At the same time, when extracting the most important predictors from the final set of rules, we generalized the results at the whole sample level and obtained common patterns. Thus, in the KIHD sample, taking no medication (prescribed or non-prescribed drugs, drugs for back or joint pain, drugs for blood pressure, prostaglandin synthetase inhibitors or pain killers, diuretics, and beta-blockers), absence of other diseases (kidney stones, colitis, chronic bronchitis, gallbladder disease, migraine, and restricted mobility), and a high standard of living defined by socioeconomic predictors (having a color television, a video recorder, and/or a dishwasher and using a car or a taxi in winter) are common traits of “easy” cases for the LLR model when predicting cardiovascular death. Conversely, “difficult” cases included subjects who had taken medication (prescribed drugs, drugs for back or joint pain, drugs for blood pressure, drugs for hypertension, prostaglandin synthetase inhibitors or pain killers, and diuretics) [32].

Although in this study, the approach proposed was applied for LLR post-analysis, it is model-independent and might be useful for better understanding of any model behavior, which is especially helpful when “black box” models are trained on high-dimensional data. Such post-analysis is also useful for “white box” models, because variables that are important for making predictions differ from variables that are informative for describing “easy” and “difficult” cases.

The main limitation of this study is that all of our findings are applicable to the KIHD cohort, but they cannot be extrapolated to other populations. To make more general conclusions, e.g., about the role of medication in predicting cardiovascular death, we need to perform the same analysis for other populations, and then check if there are rules and variables similar for different populations. Additionally, since this post-analysis was applied to a particular data-driven model with a certain prediction horizon, for other models or different prediction horizons the analysis should be re-run.

Despite these limitations, the approach proposed is useful for many epidemiological studies, as the model accuracy has been reported to be far from 100% [33,34,35,36]. “Difficult” cases should be identified so that clinicians can revise the model predictions and use their expertise when the model is likely to make a mistake. This approach, i.e., when the model is augmented with the human expertise in some cases, corresponds to the philosophy of responsible machine learning (and responsible artificial intelligence, in general), which is being discussed by researchers nowadays [37]. Thus, such post-analysis is extremely important from a practical perspective, as it supports model deployment in a “responsible” way.

## Figures and Tables

**Figure 1 healthcare-09-00792-f001:**
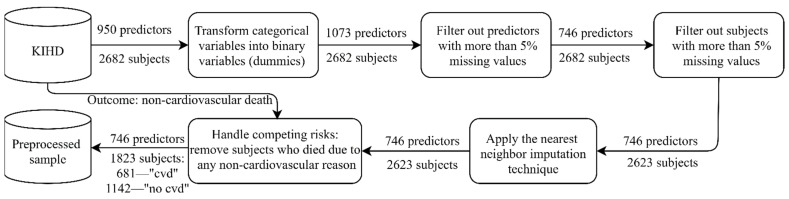
Preprocessing the sample from the Kuopio Ischemic Heart Disease Risk Factor Study (KIHD): “cvd” corresponds to “cardiovascular death by 2016”, while “no cvd” corresponds to “no cardiovascular death (alive) by 2016”. The nearest neighbor imputation technique implemented by Troyanskaya et al. was utilized [20].

**Figure 2 healthcare-09-00792-f002:**
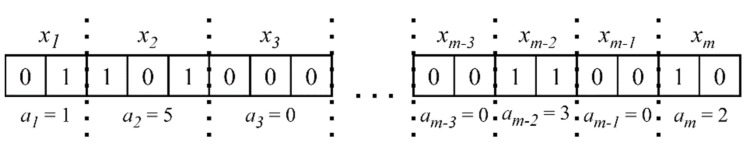
The binary representation of a rule. Parts of a binary string consecutively code the predictors’ levels or their absence in the rule.

**Figure 3 healthcare-09-00792-f003:**
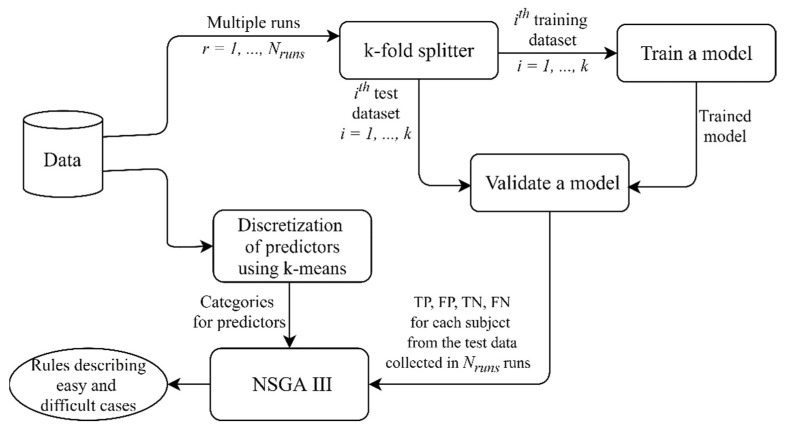
The pipeline implemented in this study. In several independent runs of cross-validation, the model performance was estimated using the test data and the results were collected for further analysis. Discretization was applied to continuous predictors to define their levels (categories) used for generating rules. NSGA III found combinations of predictors and their values describing “easy” cases, i.e., subgroups of subjects with a large number of true positive (TP) and true negative (TN) predictions, and “difficult” cases, i.e., subgroups of subjects with a large number of false positive (FP) and false negative (FN) predictions.

**Figure 4 healthcare-09-00792-f004:**
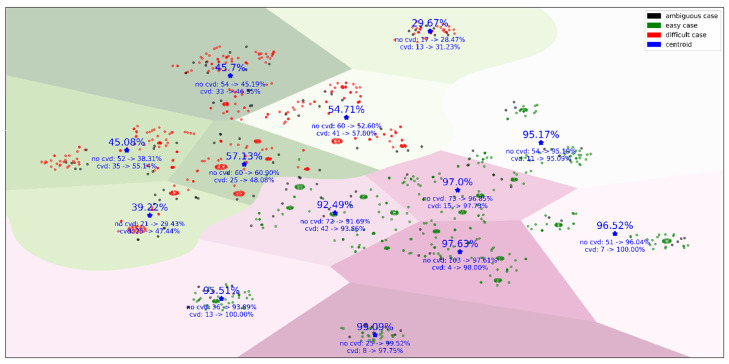
The KIHD subjects mapped from the 82-dimentional rule space onto the plane. This figure contains the subjects covered by the final set of rules. A separating line between “easy” and “difficult” cases was drawn by a support vector machine [30], and then borders between clusters were defined using the k-means method [31]. The overall model accuracy, true positive rate (TPR), and true negative rate (TNR) within clusters averaged over 50 independent runs of five-fold cross-validation, as well as the number of subjects who died by 2016 (“cvd”) and stayed alive by 2016 (“no cvd”), are given for each cluster.

**Figure 5 healthcare-09-00792-f005:**
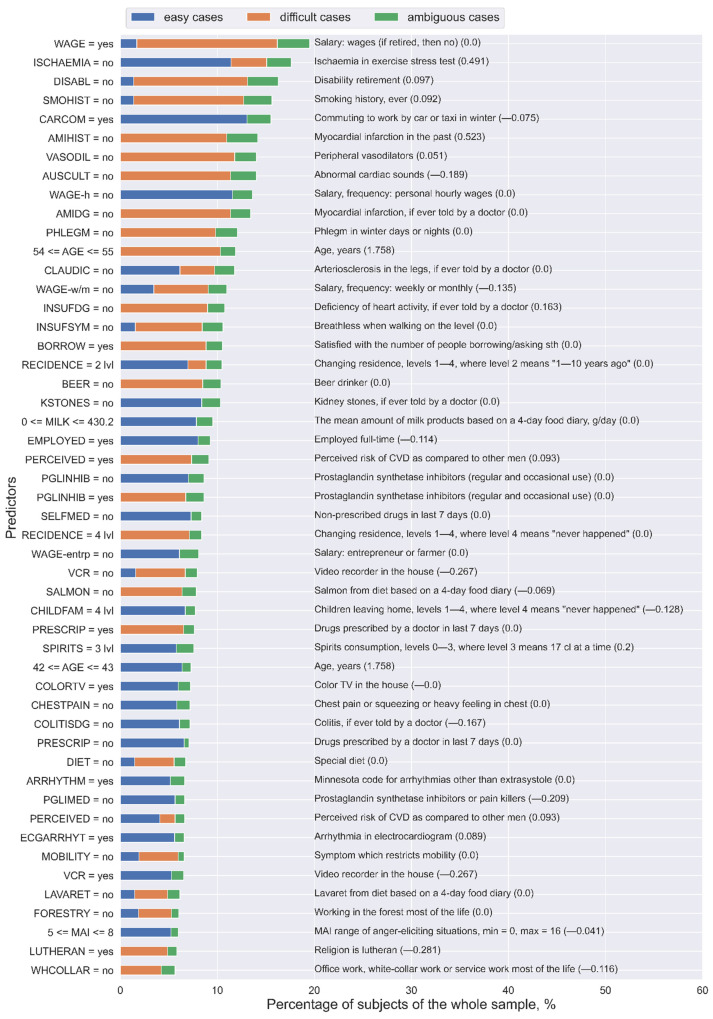
The 50 most important predictors and their values extracted from the final set of rules for the Lasso logistic regression (LLR) model using the KIHD sample. The numbers in parentheses are the weight coefficients in the LLR model for the corresponding predictors: Some of these variables were not selected as important for LLR, but they were important for analyzing “easy” and “difficult” cases.

**Table 1 healthcare-09-00792-t001:** The Non-dominated Sorting Genetic Algorithm III (NSGA III) settings used in the experiment.

Setting (Parameter) Names	Setting (Parameter) Values
Selection	Tournament selection with a tournament size of 2
Crossover	Half-uniform crossover
Mutation	Bin-flip mutation
Solution representation	Binary code → Gray code
M-objective problem	3
Outer divisions, $p_{out}$	20
Inner divisions, $p_{in}$	0
Reference points, H	$H=\left(\begin{array}{c} M+p_{out}-1\\ p_{out} \end{array}\right)+\left(\begin{array}{c} M+p_{in}-1\\ p_{in} \end{array}\right)=231$
Population size	The smallest multiple of four greater than H, i.e., 232
Generations	200
Probability distribution for initializing solutions in the starting population	$P\left(x_{j}\;\mathrm{is}\;\mathrm{not}\;\mathrm{included}\;\mathrm{in}\;\mathrm{the}\;\mathrm{rule}\right)=0.95$ $P\left(x_{j}\;\mathrm{is}\;\mathrm{included}\;\mathrm{in}\;\mathrm{the}\;\mathrm{rule}\right)=0.05$

**Table 2 healthcare-09-00792-t002:** Four categories of subjects in the KIHD sample after applying the final set of rules. The “Accuracy, %” columns represent the mean accuracy estimated in 50 runs of five-fold cross-validation. “The number of cases” columns contain the absolute number of subjects, with the percentage of the whole sample in parentheses.

	Accuracy, %			The Number of Cases			
	Easy	Difficult	Ambiguous and Non-Covered	Easy	Difficult	Ambiguous	Non-Covered
No cvd	95.73	46.76	66.55	414	264	91	373
Cvd	96.28	50.17	76.05	100	172	31	378
Overall	95.84	48.11	71.00	514 (28.20%)	436 (23.92%)	122 (6.69%)	751 (41.20%)

## Data Availability

The data presented in this study are available on request from the Institute of Public Health and Clinical Nutrition, University of Eastern Finland (A.V., T.-P.T., J.K.) The data are not publicly available due to its sensitive nature.

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
