# Peer review of "Post-Analysis of Predictive Modeling with an Epidemiological Example"

_healthcare, 2021, doi:10.3390/healthcare9070792_

Round 1

Reviewer 1 Report

This study proposed a computational framework for post-analysis of predictive modeling with an epidemiological example. I should appreciate the authors' time and patient to come up with some results. However, there are several problems that deduct from the quality of this manuscript. Below are several comments on this work.

  1. The authors are suggested to improve the quality of all figures by improving the resolutions.
  2. Please add time complexity for your method.
  3. Please add parameter sensitivity analysis for your method.
  4. Since the evolution computing method is a stochastic optimization method, the best, average, the worst, and standard deviations by the proposed method should be added.
  5. Please compare your proposed method with state-of-the-art image synthesis methods to improve your method.
  6. In Conclusion, there was no mention of the limitations of the study.
  7. Please release the sourcecodes.

Reviewer 2 Report

Review of the article: Post-analysis of predictive modeling with an epidemiological example

Any work aimed at shedding light on people's health is worthy of consideration. In this article, the authors apply a Logistic Lasso Regression to predict cardiovascular death by 2016 using the data from the examination in 1984-1989.

The effort made by the authors to develop the methodology applied to the specific case of cardiovascular death is commendable but, in my opinion, there are some flaws in the work that should be corrected prior to publication.

First, a literature review section should be included, either independently or within the Introduction. In fact, very few bibliographic references are cited. What other post-analysis methods have been previously applied? In which specific studies? What is the background in the field of medicine of studies similar to this one with other methodologies? What differences are there between those methodologies and the one proposed by the authors? What is the advantage of the methodology proposed by the authors? Has it been applied to previous studies on diseases? Etc.

The object of study is not clearly specified and should be, both in the abstract and in the Introduction.

There are no study hypotheses. Didn't the authors have them? It seems that, indeed, there were no such hypotheses and that this study is an exploratory analysis. One or the other question should be clearly stated.

The authors should also specify whether the relevant variables of the study and the rules generated can be extrapolated outside the specific sample with which they have worked. Would they be useful for a different population? It seems that not necessarily, but they should specify this. What, if any, predictive level would the proposed model have outside the sample under consideration?

The keyword “prediction of cardiovascular diseases” seems not correct. It should be “prediction of cardiovascular death”.

What does “black box” and “white box” mean? Not all readers of health topics are specialists in statistical modeling. However, the article is aimed at this audience and they should understand the authors' exposition. They should also explain what a post-analysis model is.

The authors repeat the acronym "KIHD" on page 2. It is only necessary to specify it the first time and then you can use it.

On page 2 authors should write “42-60 years old” instead of “42-60 y.o.”

The lower part of Figure 3 is not well defined. It should be reinserted.

The discussion should be broadened. Most of it is intended to repeat the explanation of the applied model. In addition, it should include the limitations of the study.

Round 2

Reviewer 1 Report

The authors have addressed all my comments.

Reviewer 2 Report

The authors have made all the changes required in the previous revision. Therefore, I consider the article to be correct for publication.
I congratulate the authors for their work and wish them the best of luck in their publication.

This manuscript is a resubmission of an earlier submission. The following is a list of the peer review reports and author responses from that submission.